# Chemo-profiling of *Purpureocillium lilacinum* and *Paecilomyces variotii* isolates using GC-MS analysis, and evaluation of their metabolites against *M. incognita*

Prashant Patidar[1], Lakshman Prasad[1]*, Sushma Sagar[1], Anil Sirohi[2], Mahender Singh Saharan[1], Mukesh Kumar Dhillon[3], Vaibhav Kumar Singh[1], Tusar Kanti Bag[1]

1 Division of Plant Pathology, ICAR-Indian Agricultural Research Institute, New Delhi, India, 2 Division of Nematology, ICAR-Indian Agricultural Research Institute, New Delhi, India, 3 Division of Entomology, ICAR-Indian Agricultural Research Institute, New Delhi, India

* laxmanprasad25@yahoo.com

## Abstract

Nematophagous fungi are the best alternatives to chemical nematicides for managing nematodes considering environmental health. In the current study, activity of metabolites from ten isolates of *Purpureocillium lilacinum* (Thom) Luangsa-ard (*Hypocreales*: *Ophiocordycipitaceae*) and two isolates of *Paecilomyces variotii* Bainier (*Eurotiales*: *Trichocomaceae*), were examined to inhibit the hatching of *Meloidogyne incognita* (Kofoid & White) Chitwood (*Tylenchida*: *Heteroderidae*) eggs. At 100%, 50%, and 25% concentrations, respectively, the culture filtrate of the isolate *P. lilacinum* 6887 prevented 97.55%, 90.52%, and 62.97% of egg hatching. Out of all the isolates, PI 6887, PI 6553, and PI 2362 showed the greatest results in the hatching inhibition experiment.Gas chromatography-mass spectrometry (GC-MS) analysis revealed a variety of nematicidal compounds from different isolates. A total of seven nematicidal compounds, including four very potent nematicidal fatty acids were found in the isolate PI 6553. Secondary metabolites of the same isolate possess the highest *M. incognita* juvenile mortality, i.e., 43.33% and 92% after 48 hrs of treatment at 100 and 200 ppm concentrations, respectively. Significant difference was observed in juvenile mortality percentage among the isolate having highest and lowest nematicidal compounds. Nematicidal fatty acids like myristic and lauric acid were found for the first time in *P. lilacinum*. Multiple vacuole-like droplets were found inside the unhatched eggs inoculated with the culture filtrate of isolate PI 6887, and also in the juveniles that perished in the ethyl acetate extract of isolate PI 6553.

## Introduction

Plant parasitic nematodes (PPNs) are potential threats to crop production. Global loss due to PPNs is estimated to be $157 billion annually [1]. Although the number of plant-parasitic nematode species identified is over 4,100, undoubtedly, the most yield-reducing group is the

**Data Availability Statement:** All relevant data are within the manuscript and its Supporting Information files.

**Funding:** The authors received no specific funding for this work.

**Competing interests:** The authors have declared that no competing interests exist.

root-knot nematode (*Meloidogyne* spp.). It ranks top in terms of agricultural damage globally [2]. Out of the 101 RKN species described, *Meloidogyne incognita* is the most economically important species in the tropical and subtropical regions [3, 4]. Along with the direct damage, an association of RKN with fungal and bacterial plant pathogens is common and results in disease complexes, causing more damage to the crops than the nematode alone [5]. Chemical nematicides are widely used to control RKNs worldwide. However, due to their adverse effects on humans, environmental health, and non-targets, the use of chemical nematicides is not feasible in the current scenario of eco-friendly crop-pest management [6]. It results in the phase-out of several popular nematicides from the market [7]. Hence, biological pesticides are becoming an important component of eco-friendly management systems [8, 9]. *Purpureocillium lilacinum* (earlier *Paecilomyces lilacinus*) is one of the various antagonists of the cyst and root-knot nematodes, which is found most effective as an egg and female parasite [10]. It is considered one of the most promising and practicable biological control agents for the management of sedentary endoparasitic PPNs [11] and has been used successfully to manage root-knot nematodes in various crop plants [10, 12–16]. Nematicidal metabolites secreted by *P. lilacinum* during infection of RKNs make its job even easier, and studies suggest that potential isolates of *P. lilacinum* are efficient producers of metabolites lethal to nematodes [17–19]. Similarly, *Paecilomyces variotii* has also been reported to produce compounds of agricultural importance, including some nematicidal ones [20].

The present study aimed at determining the efficacy of ten *P. lilacinum* and two *P. variotii* isolates against *M. incognita*, *in vitro*, as well as studying their secondary metabolites in nematode control.

## Materials and methods

### Fungal cultures

For this study, ten isolates of *Purpureocillium lilacinum* and two isolates of *Paecilomyces variotii* were obtained from the Indian Type Culture Collection (ITCC), Division of Plant Pathology, ICAR-Indian Agricultural Research Institute, New Delhi. The isolates were maintained at 25˚C on PDA media.

**Cultural and morphological identification of the fungal isolates.** Investigation on morphological characters was done using a compound microscope (ZEISS). Morphological identification of the isolates was done based on the earlier diagnostic keys and descriptions of the species [21, 22].

**Molecular identification of the fungal isolates.** The fungal DNA of all 12 isolates was extracted using the CTAB method from seven-day-old mycelia. The ITS region was amplified using the ITS primers (primer sequences: ITS1: 5′ TCCGTAGGTGAACCTGCGG 3′ and ITS4: 5′ TCCTCCGCTTATTGATATGC 3′) and PCR was performed. Confirmation of PCR amplification was done with the help of agarose gel electrophoresis. The amplified DNA was sequenced using Sanger sequencing and the obtained sequences were compared with the reference sequences available at the NCBI database to identify the species based on sequence similarities (blast.ncbi.nlm.nih.gov/). Phylogenetic analysis was performed using MEGA-X software with the Maximum Likelihood method. The Jukes-Cantor model was used for phylogenetic tree construction by keeping 1000 bootstrap replications.

### Source, molecular identification and maintainance of *Meloidogyne incognita*

The culture of *M. incognita* was obtained from Division of Nematology, ICAR-Indian Agricultural Research Institute, Pusa, New Delhi. The molecular identity of *M. incognita* was confirmed using species-specific primers. The primers used were MincF1

(5-AAAAACACGCGATAACAAAAA-3) and MincR1 (5-ATTCAAAACTTGGGGGAAAAA-3) as forward and reverse primers, respectively [23]. The Cox-1 gene primer was used as a positive control [24]. The inoculum was received in the form of egg masses from infected tomato roots. Further, *M. incognita* was maintained on a susceptible tomato (*Solanum lycopersicum* Mill. cv. Pusa ruby) grown in earthen pots filled with steam-sterilized sand and soil mixtures. For further experiments, the egg masses were collected from these infected plants.

### *In vitro* egg hatching inhibition assay

**Preparation of *M. incognita* eggs suspension.** The egg masses were separated from the roots of the infected tomato plants with the help of a needle, while eggs were separated from the egg masses using a 1% sodium hypochlorite (NaOCl) solution [25]. The eggs were passed through a 100-mesh sieve nested over 200 and 500-mesh sieves. The eggs collected over a 500-mesh sieve were gently rinsed with sterile distilled water to remove NaOCl residue and decanted in a clean sterile beaker. The egg-hatching inhibition assay was performed on freshly extracted eggs.

**Egg-hatching inhibition assay.** All twelve isolates were grown in a 500 ml Erlenmeyer flask containing 300 ml of sterilized potato dextrose broth (HiMedia), followed by incubation at 25°C for 21 days. The cell- free culture filtrate of the isolates was obtained by filtering the broth through a 0.22 μm millipore filter, and used for egg-hatching inhibition assay.

In each well of a 24-well cell culture plate, a 20 μl suspension of *M. incognita* eggs containing approximately 100 eggs was pipetted. The culture filtrate of the fungal isolates was added into the wells at three different concentrations, that is, 100%, 50%, and 25% with 4 replications each. At 50% and 25% concentrations of the culture filtrate tested, the volume was made up to 1 mL by adding sterile distilled water. Sterile distilled water was taken as negative control. The plates were incubated at 27°C in a BOD incubator after sealing them with parafilm to reduce evaporation. The number of hatched juveniles was monitored regularly, while the final counting of the hatched and unhatched eggs was done on the seventh day of inoculation using a phase contrast microscope. The egg hatching inhibition in the treatments was calculated by the formula HI % = (Total number of hatched J2 in control–Number of hatched J2 in the treatment/ Total number of hatched J2 in control) × 100 [26], which also eliminates the natural mortality in the treatments.

### GC-MS analysis

The culture filtrate of each isolate was extracted thrice with an equal volume of ethyl acetate. The aqueous phase was concentrated using a rotary evaporator (Heidolph, Germany) at reduced pressure and a temperature of 55°C. The concentrated, dried metabolites were weighed and re-dissolved in GC-MS grade ethyl acetate at the rate of 1gm per ml. The GC-MS analysis was performed as described by [27] using Shimadzu GC-MS-QP 2010 using Rtx-1 MS column (30 m × 0.25 mm; 0.25 μm i.d.) connected directly with a triple axis mass spectrometer. The injected sample volume was 1 μl. Helium (> 99.99% purity) was used as a carrier gas with a head pressure of 10 psi and flow of 0.75 mL min$^{-1}$. The oven temperature was held initially at 40°C and increased 4°C min$^{-1}$ to reach 120°C. Then the temperature increased again at the rate of 5°C min$^{-1}$ up to 200°C with a hold time of 5 min. Finally, the temperature was raised to 300°C with an increment of 10°C min$^{-1}$. The total run time was 51 min. The MS acquisition parameters were set with the ion source temperature 250°C, electron ionization 60 eV, full scan mode (50–550 AMU), transfer line temperature 280°C, and E.M voltage 1222. NIST (National Institute of Standards and Technologies) Mass Spectra Library was used for the identification of compounds by matching their mass spectra (Shimadzu Corporation, Kyoto, Japan).

### *M. incognita* Juvenile mortality assay

The juvenile mortality assay was conducted using extracted secondary metabolites of three *P. lilacinum* isolates Pl 6553, Pl 6887 and Pl 4483.

**Preparation of stock and working concentration of secondary metabolites.** The dried secondary metabolites of the isolates Pl 6553, Pl 6887 and Pl 4483 were weighed and re-dissolved in methanol (1mg/ml) to make 1000 ppm stock concentration. Further, the working suspension of 100ppm and 200ppm was prepared from stock solution for juvenile activity test.

**Preparation of *M. incognita* juveniles suspension.** Tomato plants infected with *M. incognita* were uprooted, and the roots were washed free of soil with tap water. The individual egg masses were picked from the roots with the help of a toothpick and surface sterilized with 0.1% NaOCl. The egg masses were then kept in a clean Petri plate, and sterile distilled water was added to the plate just to cover the surface of the egg masses. The plates were kept at room temperature to allow the hatching of eggs. The freshly hatched juveniles were collected using a micropipette in a clean beaker. Counting juveniles per ml was done with the help of a stereomicroscope aided by a hand tally counter. Finally, the number of juveniles was adjusted to 100/ 20 µl suspension.

**The juvenile mortality assay.** For the juvenile mortality assay, two concentrations (100 and 200 ppm) of the methanol-dissolved secondary metabolites of the three *P. lilacinum* isolates (Pl 6553, Pl 6887 and Pl 4483) were used. The 24 well cell culture plate were used for the bioassay. Each well received 100 *M. incognita* J2 (20 µl suspension), and either 10 µl (for 100 ppm concentration) or 20 µl (for 200 ppm concentration) methanol dissolved secondary metabolites of the isolates from the stock concentration of 1000 ppm. The volume was made up to 1 ml with 0.5% Tween 80. Each treatment was replicated three times.Both 10 and 20 µl of pure methanol, dissolved in 0.5% Tween 80, along with 100 *M. incognita* juveniles was taken as control. The plates were incubated at 27˚C in a BOD incubator after sealing with parafilm. After 24 and 48 hours of incubation, the live and dead juveniles were counted. The freely moving juveniles were considered alive, whereas the juveniles, being straight and did not show any movement after probing with a nematode handling needle, were considered dead [28].

### Data analysis

Completely randomized design (CRD) was followed as experimental design for both egg-hatching inhibition and juvenile mortality experiments. All percentage data were first transformed into arcsine value and then used for two-way Analysis of Variance (ANOVA) for hatching inhibition assay and three-way Analysis of Variance (ANOVA) for juvenile mortality assay. The post hoc multiple comparisons were done at a 5% significant level by Duncan's Multiple Range Test (DMRT). All the analysis was carried out using OPSTAT software (http://14.139.232.166/opstat/).

## Results

### Cultural and morphological identification of the fungal isolates

The ten isolates of *P. lilacinum* were confirmed as *Purpureocillium lilacinum* using the reference keys [22]. The colony colour on PDA media for all the isolates of *P. lilacinum* was lilac to vinaceous, which is typical of this fungus. Whereas morphological investigations revealed the production of phialides with long tapering necks, and ellipsoidal to fusiform conidia, varied from 2.6–3.40 × 2–2.8 µm in size. Both isolates of The cultural and morphological characters of both *P. variotii* isolates were matched with the keys [22, 29] as they produced a white fluffy colony, which turns yellowish brown or sand colored during conidiogenesis. They also

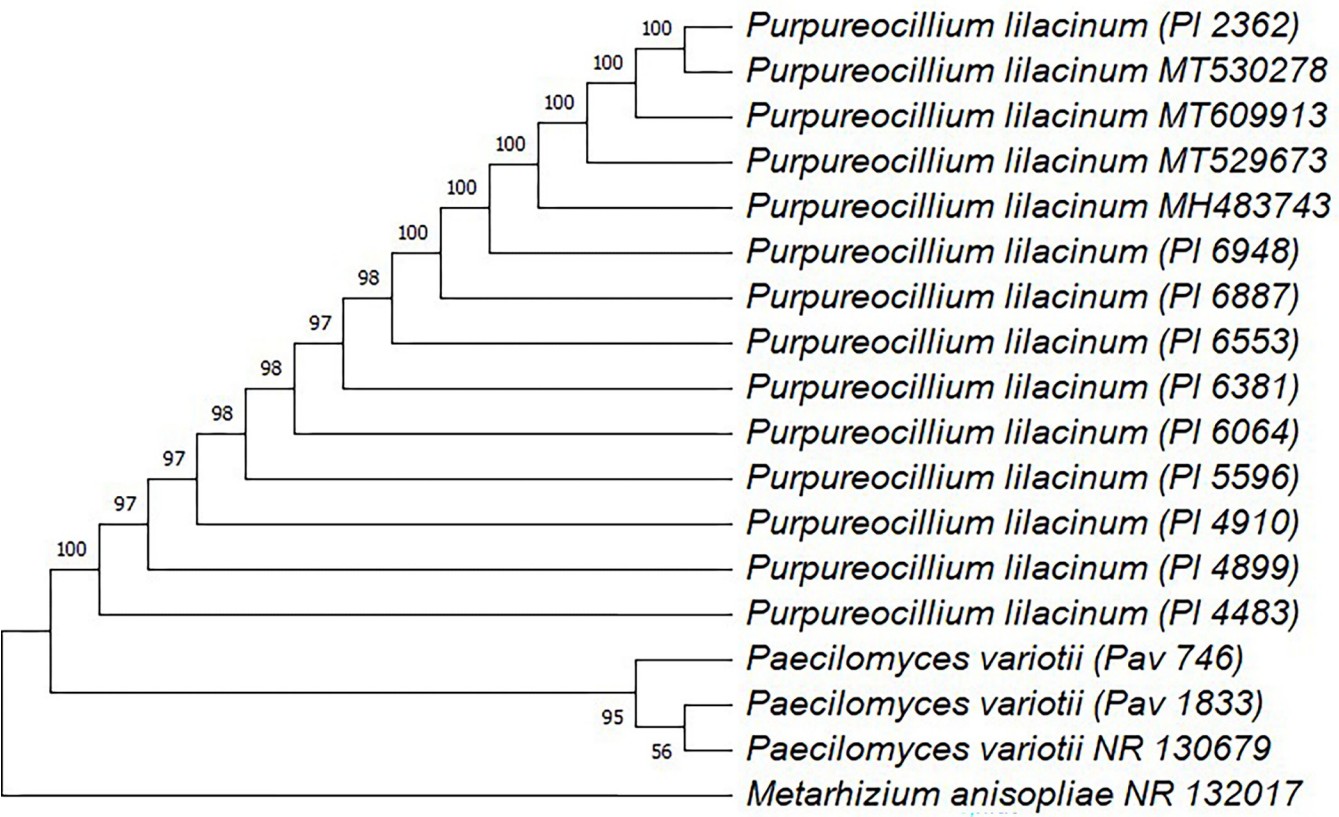

**Fig 1. Phylogenetic relationship between isolates of *Purpureocillium lilacinum* and *Paecilomyces variotii* based on ITS sequences constructed using the maximum likelihood method.** The percentage of trees in which the associated taxa clustered together is shown next to the branches and are based on 1000 bootsrap replications.

produced fusoid to broadly ellipsoid conidia, varied from 3.3–4.8 × 2–2.4 μm in size with visible multiple nuclei inside. Chlamydospores were observed only in one isolate (Pav 746), among the two isolates of *P. variotii* studied.

## Molecular identification of the fungal isolates

After sequencing of PCR products of all the 12 isolates, the obtained sequences were BLAST at the NCBI database, which confirms ten *P. lilacinum* isolates and two *P. variotii* isolates. The GenBank accession number of all the isolates is mentioned in the S1 Table. The phylogenetic tree was constructed using the maximum likelihood method, which shows the close relatedness of *P. lilacinum* and *P. variotii* isolates with already available sequences of *P. lilacinum* and *P. variotii*, respectively, on the NCBI database (Fig 1).

## Molecular identity of *M. incognita*

Molecular identity of *M. incognita* was confirmed using species-specific primers [23]. It produced about 150 bp amplicon (Fig 2). The Cox-1 gene was used as a positive control [24].

## Egg-Hatching inhibition assay

Significant variation in hatching inhibition was found among the treatments as compared to the control (Table 1). The isolate, Pl 6887 was the best-performing isolate, as the highest

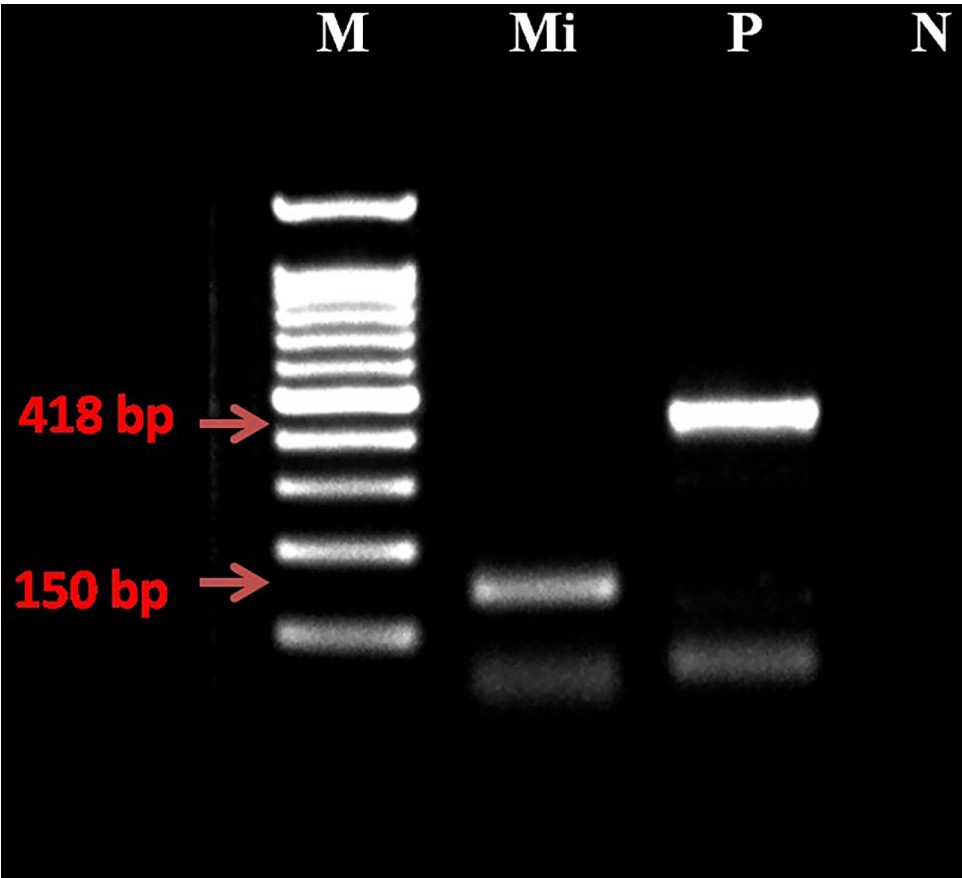

**Fig 2.** *Meloidogyne incognita* **molecular confirmation.** PCR products with MincF1/R1 primer set. M: 100 bp ladder, Mi: 150 bp, (MincF1 and Minc R1), P: Positive control- Cox-1 gene, 418 bp (JB3 and JB5 primers) and N: No template control.

hatching inhibition (97.55%, 90.52%, and 62.97%) was achieved at 100%, 50%, and 25% concentration, respectively. It was the only isolate where more than 50% hatching inhibition was observed at the lowest concentration (25%) tested. However, hatching inhibition in the culture filtrate of the isolates Pl 6553 was comparable to Pl 6887 at 100% and 50% concentrations. The culture filtrate of the *Paecilomyces variotii* isolate Pav 1833, showed 90.52%, 56.88%, and 28.74% hatching inhibition at 100, 50, and 25% concentrations, respectively. The result obtained was better than three isolates of *P. lilacinum* (Pl 4483, Pl 5596 and Pl 6948). However, hatching inhibition in another *Paecilomyces variotii* isolate (Pav 746) was the lowest among all the isolates at 100% and 25% concentrations. Unhatched eggs of *M. incognita* from the culture filtrate of the isolates, Pl 6887 and Pl 6553, were deformed and vacuolated (Fig 3) when observed under phase contrast microscope at higher magnification.

## Chemo-profiling analysis using Gas Chromatography Mass Spectroscopy

GC-MS analysis revealed a large number of bioactive compounds from all the 12 isolates (S1–S12 Figs). However, as of our interest, the compounds found as nematicidal in earlier studies [17–19, 30–35] were shortlisted and mentioned with their retention time and area in Table 2. Identification of compounds were done based on NIST database. The number of nematicidal compounds varied among the isolates. The highest number (seven) of nematicidal compounds

**Table 1. List of isolates *Paecilomyces variotii* and *Purpureocillium lilacinum* and effect of their culture filtrates on *M. incognita* egg hatching.**

| *Treatments | Name of Isolates | Hatching inhibition (%) | | | Means |
|---|---|---|---|---|---|
| | | 100% conc. | 50% conc. | 25% conc. | |
| T1 | **Pav 746 | †84.40 ‡(66.78)[f****] | 48.93 (44.39)[g] | 10.71 (19.03)[g] | 48.01(43.38) |
| T2 | Pav 1833 | 90.52 (72.14)[cd] | 56.88 (48.96)[f] | 28.75 (32.4)[de] | 58.71(51.14) |
| T3 | ***Pl 2362 | 95.11 (77.35)[b] | 73.70 (59.16)[cd] | 37.92 (37.99)[c] | 68.91(58.14) |
| T4 | Pl 4483 | 87.16 (69.03)[ef] | 42.51 (40.68)[h] | 12.54 (20.6)[g] | 47.40(43.41) |
| T5 | Pl 4899 | 92.05 (73.68)[c] | 75.54 (60.36)[c] | 35.17 (36.36)[c] | 67.58(56.77) |
| T6 | Pl 4910 | 91.13 (72.7)[c] | 67.28 (55.12)[e] | 30.89 (33.75)[d] | 63.09(53.83) |
| T7 | Pl 5596 | 85.63 (67.73)[ef] | 19.57 (26.22)[i] | 11.93 (20.11)[g] | 39.04(38.01) |
| T8 | Pl 6064 | 94.19 (76.1)[b] | 70.95 (57.39)[d] | 17.74 (24.85)[f] | 60.95(52.75) |
| T9 | Pl 6381 | 94.80 (76.92)[b] | 80.73 (63.97)[b] | 17.13 (24.42)[f] | 64.22(55.08) |
| T10 | Pl 6553 | 96.94 (79.98)[a] | 89.30 (70.91)[a] | 44.96 (42.1)[b] | 77.06(64.30) |
| T11 | Pl 6887 | 97.56 (81.31)[a] | 90.52 (72.09)[a] | 62.97 (52.53)[a] | 83.68(68.61) |
| T12 | Pl 6948 | 88.38 (70.1)[de] | 40.67 (39.62)[h] | 26.00 (30.63)[e] | 51.68(46.76) |
| T13 | Control | 0.00 (0)[g] | 0.00 (0)[j] | 0.00 (0)[h] | 0.00(00.00) |
| CD @p <0.05 | A (Isolates) | 1.300 | | | |
| | B(Concentration) | 0.625 | | | |
| | AxB | 2.252 | | | |

*Fungal culture filtrate and 100 eggs of *M. incognita*

**Pav = *Paecilomyces variotii*

***Pl = *Purpureocillium lilacinum*.

† Mean of 4 replications in percent; ‡ Arcsine transformation values in parentheses.

****Values in the same column, followed by common letter(s) are statistically not different as per Duncan's multiple range test at 5% level of significance (p<0.05).

were found in the ethyl acetate fraction of Pl 6553 isolate, followed by five in Pl 6887 (Table 2). Nematicidal fatty acids, namely, lauric (dodecanoic acid), myristic (tetradecanoic acid), stearic (octadecanoic acid) and butyric (butanoic acid) acid were found in Pl 6553 isolate, along with other nematicidal compounds like benzene acetic acid, 4-hydroxy benzene ethanol and benzoic acid. Only one reported nematicidal compound found in Pl 4483 isolate i.e., palmitic acid

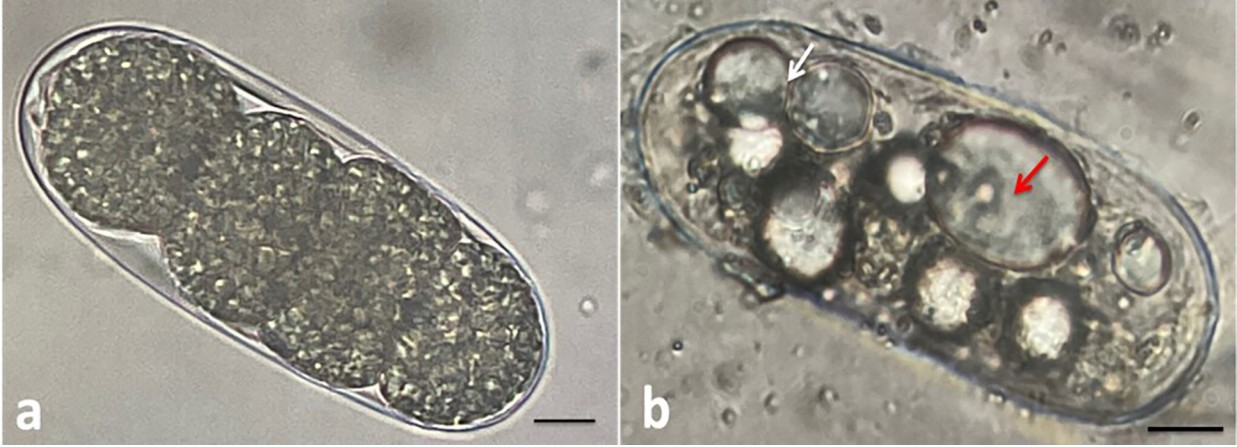

**Fig 3. Effect of culture filtrate of *P. lilacinum* on *M. incognita* eggs.** (a) Control, where the egg is under embryogenesis, and (b) egg with a giant vacuole (red arrow), two vacuoles fusing (white arrow) and multiple variously sized vacuoles like droplets in the culture filtrate of *P. lilacinum* isolate Pl 6887, at 7th day of incubation. Scale bars of a and b were 50 μm.

**Table 2. List of nematicidal compounds identified using GC-MS analysis in *Paecilomyces variotii* and *P. lilacinum* isolates.**

| Nematicidal compounds | Molecular formula | RT* in minute | Retention area (%) | | | | | | | | | | | | References**** |
| --- | --- | --- | --- | --- | --- | --- | --- | --- | --- | --- | --- | --- | --- | --- | --- |
| | | | Pav** 746 | Pav 1833 | Pl*** 2362 | Pl 4483 | Pl 4899 | Pl 4910 | Pl 5596 | Pl 6064 | Pl 6381 | Pl 6553 | Pl 6887 | Pl 6948 | |
| Phenyl ethyl alcohol | $C_8H_{10}O$ | 13.76 | 25.40 | 20.93 | 1.56 | — | — | — | 1.78 | — | — | — | 3.66 | — | [19] |
| Acetic acid | $CH_3COOH$ | 18.61 | — | — | — | — | — | — | — | — | — | 5.32 | — | — | [30] |
| Benzene acetic acid | $C_8H_8O_2$ | 19.54 | — | 5.44 | 4.11 | — | — | — | — | — | — | 1.25 | — | — | [19] |
| 4-hydroxy benzene ethanol | $C_8H_{12}O_2$ | 24.53 | 15.47 | 29.27 | — | — | 3.52 | — | 3.63 | — | — | 1.75 | 6.40 | — | [19] |
| Lauric acid | $C_{12}H_{24}O_2$ | 27.71 | — | — | — | — | — | — | — | 0.30 | — | 0.11 | — | — | [31] |
| Benzoic acid | $C_7H_6O_2$ | 33.05 | 0.18 | — | 0.46 | — | 0.37 | — | 0.24 | 0.09 | — | 0.14 | — | — | [19] |
| Myristic acid | $C_{14}H_{28}O_2$ | 32.25 | — | — | — | — | — | — | — | 0.18 | — | 0.16 | — | — | [31, 33] |
| Palmitic acid | $C_{16}H_{32}O_2$ | 36.45 | 0.77 | 1.49 | 1.50 | 1.60 | 1.89 | 0.76 | — | 1.53 | 3.98 | — | 1.65 | 1.43 | [18, 31, 33, 34] |
| Oleic acid | $C_{18}H_{34}O_2$ | 41.10 | — | — | — | — | 0.05 | 0.14 | 0.04 | — | — | — | — | 0.11 | [18, 31, 33, 34, 36] |
| Linoleic acid | $C_{18}H_{32}O_2$ | 41.47 | — | — | — | — | — | — | — | — | 1.62 | — | — | 0.62 | [18, 31, 33, 34] |
| Stearic acid | $C_{18}H_{36}O_2$ | 41.77 | — | — | — | — | — | — | 0.09 | — | 0.20 | 0.22 | 0.15 | — | [33, 34, 37–39] |
| Butyric acid | $C_4H_8O_2$ | 48.95 | — | — | — | — | — | 0.48 | 0.74 | 0.85 | — | 0.43 | — | — | [33, 38] |

**Note**: '—' means not detected

\* the compounds are listed based on their retention time in minute

\*\**Paecilomyces variotii*

\*\*\* *Purpureocillium lilacinum*

\*\*\*\*References where these compounds are reported as nematicidal and not necessarily secreted by *P. lilacinum* or *Paecilomyces variorti*

(hexadecanoic acid). No nematicidal fatty acid, other than palmitic acid, was found in the ethyl acetate extract of *P. variotii* isolates.

## The juvenile mortality assay

The juvenile mortality assay was conducted with three isolates of *P. lilacinum*. Two isolates, Pl 6887 and Pl 6553 showed higher hatching inhibition as well as higher number of nematicidal compounds in GC-MS analysis. While the isolate Pl 4483 showed to secrete only one nematicidal compound, were included in the juvenile mortality assay to compare with Pl 6887 and Pl 6553. The ethyl acetate extract of all three *P. lilacinum* isolates showed differences in the mortality of *M. incognita* juveniles (Table 3). The isolate Pl 6553 showed the highest mortality of the juveniles i.e., 11.66% and 43.33% at 100 ppm, and 25% and 92% at 200 ppm concentrations, after 24 and 48 hours of treatment, respectively. While the lowest juvenile mortality was observed with Pl 4483 i.e., 4% and 10% at 100 ppm, and 10% and 18.66% at 200 ppm, after 24 and 48 hours of treatment, respectively. The juvenile mortality observed in the Pl 6887 isolate was in between the two other isolates (Pl 6553 and Pl 6887) and so was the number of nematicidal compounds. The juveniles in control were observed freely moving and lacking any vacuoles inside the body (Fig 4A), while multiple vacuoles were observed in the juveniles after 48 hours of incubation with the metabolites of Pl 6553 isolate (Fig 4B).

## Discussion

Chemical nematicides are still a top priority among growers for crop protection against nematodes even after the restriction and lack of availability. However, interest in the biological

**Table 3. Effect of *P. lilacinum* metabolites on *M. incognita* juvenile mortality.**

| Treatments | Isolates | % Juvenile mortality | | | |
|---|---|---|---|---|---|
| | | 100 ppm concentration | | 200 ppm concentration | |
| | | 24 h | 48 h | 24 h | 48 h |
| T1 | *Pl 6553 | †11.66 (‡19.94) [a]*** | 43.33 (41.17)[a] | 25.00 (29.99)[a] | 92.00 (73.59)[a] |
| T2 | Pl 6887 | 8.66 (17.12) [b] | 27.33 (31.52)[b] | 14.66 (22.50)[b] | 57.00 (49.02)[b] |
| T3 | Pl 4483 | 4.00 (11.48) [c] | 10.00 (18.42)[c] | 10.00 (18.42)[c] | 18.66 (25.52)[c] |
| T4 | **Control | 2.33 (8.74) [d] | 5.66 (13.76)[d] | 3.33 (10.49)[d] | 7.33 (15.70)[d] |
| CD @p <0.05 | A (Isolates) | 1.027 | | | |
| | B (Concentration) | 0.726 | | | |
| | C (Time interval) | 0.726 | | | |
| | AxBxC | 2.054 | | | |

*Pl = *Purureocillium lilacinum*

**Control = (1% Methanol in 0.5% tween 80 + 100 J2 for 100 ppm, and 2% Methanol in 0.5% tween 80 + 100 J2 for 200 ppm concentration was taken as control).

†† Mean of 4 replications in percent; ‡ Arcsine transformation values in parentheses.

***Values in the same column, followed by a common letter, are statistically not different as per Duncan's multiple range test at 5% level of significance (p<0.05).

control of nematodes has increased in recent years, and studies suggest natural products can replace the hazardous chemical nematicides [33, 40]. Nematophagous fungi have enough potential to suppress the nematode population in the field, without causing any adverse effects on the environment [10].

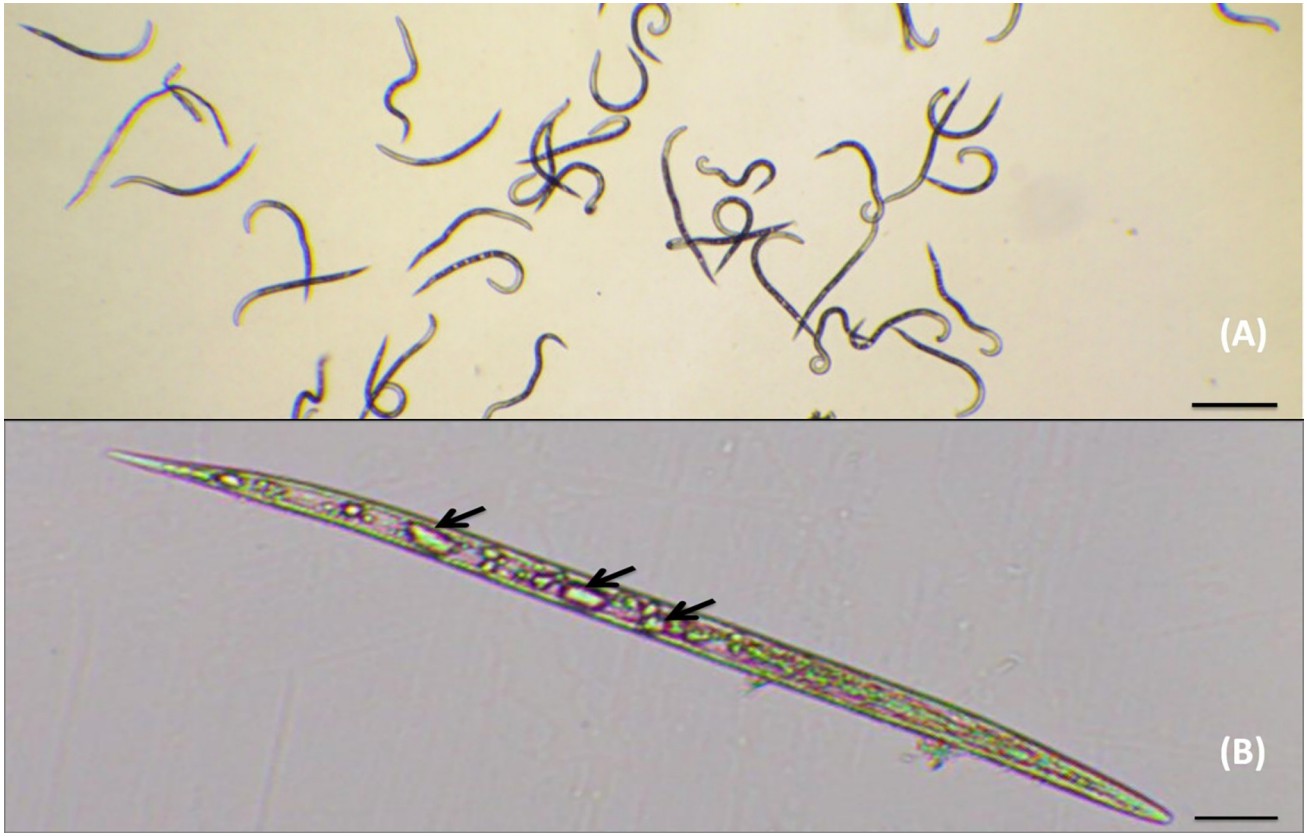

**Fig 4.** *M. incognita* juvenile after 48 hours of incubation in (A) control where juveniles were freely moving (scale bar 200 μm) (B) a dead juvenile in metabolites of *P. lilacinum* (Pl 6553) with vacuoles-like droplets (black arrow) after 48 hours of incubation (scale bar 50 μm).

In this study, the nematicidal potential of ten *P. lilacinum* and two *P. variotii* isolates was studied against *M. incognita*. Our study showed that the 21-day-old culture filtrate of *P. lilacinum* isolates can inhibit the hatching of *M. incognita* eggs up to varying extent. Eggs-hatching inhibition up to 97.55%, 90.52% and 62.97% was obtained with the culture filtrate of Pl 6887 isolate at 100%, 50% and 25% concentrations, respectively. However, in a study of [41], hatching inhibition was quite lower with the 8-day-old culture filtrate of the same isolate. It suggests that the accumulation of compounds detrimental to nematode egg hatching is greater as the age of fungal culture increases in broth. Similar kind of results were found by [42], where the 30-day-old culture filtrate of *P. lilacinum* completely inhibited *M. incognita* egg-hatching. Similarly, many studies support that the culture filtrate of *P. lilacinum* is effective in inhibiting egg hatching of various root-knot nematode species [32, 43, 44]. In a study [45] egg-hatching inhibition of *M. incognita* by culture filtrate of *P. variotii* isolates was reported. However, it was not better than *P. lilacinum* isolates. In our study, *P. variotii* isolates, Pav 1833, showed 90.58%, 56.88%, and 28.74% hatching inhibition at 100, 50 and 25% concentrations, respectively, which was found better than the results obtained with three isolates of *P. lilacinum* (Pl 4483, Pl 5596 and Pl 6948). The culture filtrate of the *P. lilacinum* isolates, especially, Pl 6887 and Pl 6553 was found toxic to the *M. incognita* eggs. The eggs were found vacuolated and deformed. Similar morphological changes in the eggs due to exposure to the culture filtrate of *P. lilacinum* are reported earlier [44].

In our study, we found that the ethyl acetate extract of *P. lilacinum* isolates was toxic to *M. incognita* juveniles. Earlier report [46] proved the higher toxicity of ethyl acetate extract of *P. lilacinum* to *M. javanica* juveniles. The per cent mortality achieved with Pl 6553 isolate was 43.33% and 92% after 48 hours of exposure at 100 and 200 ppm concentrations, respectively. [31], screened nematicidal activities of fatty acids produced in cultures of two Basidiomycetes viz., *Pleurotus pulmonarius* and *Hericium coralloides*. It was found that the LD50 value of linoleic and myristic acid against *Caenorhabditis elegans* was 5 ppm, 25 ppm for lauric, palmitic and oleic acid, and 50 ppm for stearic acid. Their study highlighted the importance of saturated and unsaturated fatty acid mixtures produced by nematophagous fungi and their contribution to nematicidal activity. [33], determined the nematicidal property of fatty acids (butyric, caproic, caprylic, capric, lauric, myristic, palmitic, oleic and linoleic acid) against *M. incognita*. It was found that the fatty acids were effective in inhibiting egg-hatching, killing second-stage juveniles and overall suppression *M. incognita* population, both in *in-vivo* and *in vitro* conditions. However, those fatty acids were not obtained from *P. lilacinum*. In our study, GC-MS analysis revealed a total of seven nematicidal compounds from Pl 6553 isolate, including four potent nematicidal fatty acids, namely, lauric, myristic, stearic, and butyric acid, with a per cent area of 0.11, 0.16, 0.22, and 0.43, respectively. [17], identified linoleic acid and oleic acid from cultures of *P. lilacinum* isolates. Production of palmitic, stearic, oleic and linoleic acid was also reported earlier from *P. lilacinum* [18, 34]. The nematicidal potential of butyric acid was described earlier [33, 37–39] with the majority of the reports suggesting butyric acid is effective against *Meloidogyne* spp. However, there are fewer reports where it was secreted by *P. lilacinum* [34]. Butyric acid was present in four isolates of *P. lilacinum* we studied, namely, Pl 4910 (0.48), Pl 5596 (0.74), Pl 6064 (0.85), and Pl 6553 (0.43).

Our study is the first report on the production of lauric and myristic acid by *P. lilacinum* (isolates Pl 6553 and Pl 6064). However, both lauric and myristic acid were reported earlier from the toxin-producing Basidiomycetes [31]. The juvenile mortality achieved with the metabolites of Pl 6553 was highest followed by Pl 6887, and least with Pl 4483. GC-MS analysis showed 7, 5 and 1 nematicidal compounds from these isolates, respectively. It suggests that the higher juvenile mortality of Pl 6553 metabolites was compound effect of the nematicidal compounds, which were present in higher number as compared to other isolates two isolates, Pl 6887 and Pl 4483. The toxic effect of nematicidal compounds secreted by Pl 6553 was visible as

the formation of multiple vacuoles inside the dead juveniles, the phenomenon termed methuosis [47]. Palmitic acid was found in every isolate studied, except for Pl 5596 and Pl 6553. Whereas, it was the only nematicidal fatty acid found in both the *P. variotii* isolates. It may conclude that *P. variotii* is a less efficient producer of nematicidal fatty acids contrary to *P. lilacinum*. Acetic acid with a high per cent area (5.32%), was only found in Pl 6887, which is also described as nematicidal earlier [30, 48].

Sharma et al. [19] found a fraction of 20 compounds, with 5 major ones, namely, 2-ethyl butyric acid, Phenyl ethyl alcohol, Benzoic acid, Benzene acetic acid and 3,5-Di-t-butyl phenol, which gave 94.6% egg hatching inhibition, and 62% juvenile mortality against *M. incognita*. However, the nematicidal effect of those five compounds was not determined individually. These compounds, except for 2-ethyl butyric acid and 3,5-Di-t-butyl phenol, were also detected in our study from different isolates. Comparatively, the per cent area of some of these compounds was much higher in the isolates of *P. variotii* than in *P. lilacinum*. However, the hatching inhibition obtained from the culture filtrate of *P. lilacinum* isolates was comparatively higher than *P. variotii* isolates. It suggests that there may be some other effective compounds playing a role in inhibiting egg hatching that need to be identified.

Our study, supported by the previous reports [17, 31, 34, 49], shows that higher hatching inhibition and mortality of juveniles, and quantity, as well as quality of the nematicidal compounds present can be an excellent indicator of the nematicidal potential of a fungal biocontrol agents. The *P. lilacinum* isolate Pl 6553, performed best in all such aspects including toxicity of its metabolites on eggs and juveniles. It also suggests that nematicidal compounds, derived from microbes, can be inexpensive alternatives to the chemical nematicides apart from being eco-friendly and selective. We found significant variation in the hatching inhibition obtained from the culture filtrate of *P. lilacinum* and *Paecilomyces variotii* isolates. Along with that, GC-MS analysis of metabolites of these isolates revealed a range of nematicidal compounds, while a couple of them are reported for the first time. The presence of a higher number of nematicidal compounds correlated with the mortality of *M. incognita* juveniles.

## Conclusion

The isolates of *P. lilacinum* varied in the production of nematicidal compounds and the presence of these compounds in higher numbers plays a significant role in an overall reduction of the root-knot nematode by inhibiting egg-hatching and/or killing juveniles. We found that the culture filtrate of the isolates Pl 6887 and Pl 6553 was highly effective in inhibiting *M. incognita* egg hatching. Along with that, Pl 6553 secreted compounds that are highly detrimental to second-stage juveniles. Our study demonstrates that the presence of nematicidal secondary metabolites in the solvent affects the survival of nematodes, and higher juvenile mortality can be achieved when more nematicidal compounds are produced by the perticular isolate. While Pl 6887 is already a commercialized isolate in India, Pl 6553 is a potential candidate for further research and field trials, with its diverse range of nematicidal compounds.

## Supporting information

**S1 Table. GenBank accession number of *P. lilacinum* and *P. variotii* isolates used in the study.**
(PDF)

**S1 Fig. GC-MS chromatogram of secondary metabolites from *P. variotii* 746 isolate (PNG).**
(PNG)

**S2 Fig. GC-MS chromatogram of secondary metabolites from *P. variotii* 1833 isolate (PNG).**
(PNG)

**S3 Fig. GC-MS chromatogram of secondary metabolites from *P. lilacinum* 2362 isolate (PNG).**
(PNG)

**S4 Fig. GC-MS chromatogram of secondary metabolites from *P. lilacinum* 4483 isolate (PNG).**
(PNG)

**S5 Fig. GC-MS chromatogram of secondary metabolites from *P. lilacinum* 4899 isolate (PNG).**
(PNG)

**S6 Fig. GC-MS chromatogram of secondary metabolites from *P. lilacinum* 4910 isolate (PNG).**
(PNG)

**S7 Fig. GC-MS chromatogram of secondary metabolites from *P. lilacinum* 5596 isolate (PNG).**
(PNG)

**S8 Fig. GC-MS chromatogram of secondary metabolites from *P. lilacinum* 6064 isolate (PNG).**
(PNG)

**S9 Fig. GC-MS chromatogram of secondary metabolites from *P. lilacinum* 6381 isolate (PNG).**
(PNG)

**S10 Fig. GC-MS chromatogram of secondary metabolites from *P. lilacinum* 6553 isolate (PNG).**
(PNG)

**S11 Fig. GC-MS chromatogram of secondary metabolites from *P. lilacinum* 6887 isolate (PNG).**
(PNG)

**S12 Fig. GC-MS chromatogram of secondary metabolites from *P. lilacinum* 6948 isolate (PNG).**
(PNG)

## Acknowledgments

The authors wants to thank Head, Division of Plant Pathology and In-charge, Indian Type Culture Collection, IARI, New Delhi for providing the fungal isolates.

## Author Contributions

**Conceptualization:** Lakshman Prasad, Mahender Singh Saharan.

**Data curation:** Prashant Patidar, Lakshman Prasad, Mukesh Kumar Dhillon.

**Formal analysis:** Prashant Patidar, Lakshman Prasad, Sushma Sagar, Mahender Singh Saharan, Mukesh Kumar Dhillon.

**Investigation:** Prashant Patidar.

**Methodology:** Prashant Patidar, Lakshman Prasad, Sushma Sagar, Anil Sirohi, Mukesh Kumar Dhillon.

**Project administration:** Prashant Patidar.

**Resources:** Prashant Patidar, Lakshman Prasad, Anil Sirohi.

**Software:** Prashant Patidar, Mukesh Kumar Dhillon.

**Supervision:** Lakshman Prasad, Anil Sirohi, Mahender Singh Saharan, Vaibhav Kumar Singh, Tusar Kanti Bag.

**Validation:** Prashant Patidar, Lakshman Prasad, Anil Sirohi, Mukesh Kumar Dhillon.

**Visualization:** Prashant Patidar, Anil Sirohi, Mahender Singh Saharan, Mukesh Kumar Dhillon, Tusar Kanti Bag.

**Writing – original draft:** Prashant Patidar, Lakshman Prasad, Sushma Sagar, Anil Sirohi.

**Writing – review & editing:** Lakshman Prasad, Mahender Singh Saharan.

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
