## [Decision Letter · Decision Letter 0]

10 Oct 2023

PONE-D-23-25925Chemo-profiling of Purpureocillium lilacinum and Paecilomyces variotii isolates using GC-MS, and bio-efficacy of their metabolites against Meloidogyne incognitaPLOS ONE

Dear Dr. Prasad,

Thank you for submitting your manuscript to PLOS ONE. After careful consideration, we feel that it has merit but does not fully meet PLOS ONE’s publication criteria as it currently stands. Therefore, we invite you to submit a revised version of the manuscript that addresses the points raised during the review process.

We look forward to receiving your revised manuscript.

Kind regards,

Ebrahim Shokoohi

Academic Editor

PLOS ONE

Journal Requirements:

**Additional Editor Comments:**

Dear Authors

Your paper contains such good information; however, it needs a significant revision.

1-The English style must be revised by a professional native English speaker.

2-GC result must be provided in material and methods, result and discussion by details.

3-Molecular identification of Meloidogyne incognita must be provided.

4-The references must be in line with the PloS One style.

Reviewers' comments:

Reviewer's Responses to Questions

**Comments to the Author**

1. Is the manuscript technically sound, and do the data support the conclusions?

Reviewer #1: Partly

2. Has the statistical analysis been performed appropriately and rigorously? 

Reviewer #1: Yes

3. Have the authors made all data underlying the findings in their manuscript fully available?

Reviewer #1: No

4. Is the manuscript presented in an intelligible fashion and written in standard English?

Reviewer #1: No

5. Review Comments to the Author

Reviewer #1: The authors conducted an integrating research's on Meloidogyne. However, the writing style needs a huge revision to be accepted. The comments and revisions are in the attached file. Some issues include:

1-What is the situation of the laboratory for conducting mortality and inhibition of egg for Meloidogyne?

2-Figure 1 for juvenile of Meloidogyne is lack of quality

3-The graphs for GC was not provided

4-The source of Meloidogyne is not clear

5-The experimental design is not clear

6-The molecular identification of Meloidogyne is not provided

7-Why authors have used Duncan for analysis?

6. PLOS authors have the option to publish the peer review history of their article (what does this mean?). If published, this will include your full peer review and any attached files.

Reviewer #1: No

---

## [Author Response · Author response to Decision Letter 0]

25 Nov 2023

Response to Reviewer #1: 

The comments and revisions are in the attached file. Some issues include:

Query 1-What is the situation of the laboratory for conducting mortality and inhibition of egg for Meloidogyne?

Reply 1. The laboratories where experiments were conducted are in Division of Nematology, and Division of Plant Pathology, ICAR- IARI, New Delhi. The laboratories are well established, well maintained and have all the facilities required to conduct juvenile mortality and hatching inhibition assay. The mass culture of M. incognita was developed under glasshouse controlled conditions on tomato. The experiments were conducted under the supervision of experts in the field of nematology and plant pathology. 

Query 2-Figure 1 for juvenile of Meloidogyne is lack of quality.

Ans 2. We have included the figures of better quality for juvenile of Meloidogyne.

Query 3-The graphs for GC was not provided.

Reply 3. The GC chromatogram are included in the supporting figure (Fig S1 to Fig S12).

Query 4-The source of Meloidogyne is not clear.

Reply 4. The Meloidogyne incognita was obtained from Division of Nematology, ICAR-IARI, New Delhi. 

Query 5-The experimental design is not clear

Reply 5. The complete randomized design was followed.

Query 6-The molecular identification of Meloidogyne is not provided.

Reply 6. The molecular identification of Meloidogyne incognita with figure is now provided in the revised manuscript.

Query 7-Why authors have used Duncan for analysis?

Reply 7. Duncan multiple range test is one of the most common method to compare treatment means, especially when difference are larger among treatments, as our data in hatching inhibition and juvenile mortality assay possesses larger difference in their means. Along with that, DMRT is most commonly used method to compare means in biological experiments. This is the reason we used Duncan for analysis.

Response to additional Editor Comments:

Your paper contains such good information; however, it needs a significant revision.

1-The English style must be revised by a professional native English speaker.

Reply: We have tried our best to improve English style with English professional.

2-GC result must be provided in material and methods, result and discussion by details.

Reply: GC-MS related all information included in material and methods, results and discussion in details.

3-Molecular identification of Meloidogyne incognita must be provided.

Reply: The molecular identification of Meloidogyne incognita with figure is now provided in the revised manuscript.

4-The references must be in line with the PloS One style.

Reply: References has been corrected as per Plos One style.

---

## [Editor Report · Decision Letter 1]

1 Dec 2023

PONE-D-23-25925R1Chemo-profiling of Purpureocillium lilacinum and Paecilomyces variotii isolates using GC-MS analysis, and evaluation of their metabolites against M. incognitaPLOS ONE

Dear Dr. Prasad,

Thank you for submitting your manuscript to PLOS ONE. After careful consideration, we feel that it has merit but does not fully meet PLOS ONE’s publication criteria as it currently stands. Therefore, we invite you to submit a revised version of the manuscript that addresses the points raised during the review process.

We look forward to receiving your revised manuscript.

Kind regards,

Ebrahim Shokoohi

Academic Editor

PLOS ONE

Journal Requirements:

Additional Editor Comments:

The authors improved the paper and addressed the raised questions. However, a minor change should be applied to the manuscript:

1-The phylogenetic tree, Figure 2, number 0.00 and other with decimal must be deleted from the tree. Keep only percentage. In addition, the percentage must be in the right place, not to cross the lines. Branch length no need to given. The legend for the Figure 2 then accordingly must be corrected.

Reviewers' comments:

The ms improved and the questions answered.

---

## [Author Response · Author response to Decision Letter 1]

5 Jan 2024

Regarding reference list

The reference list is reviewed and corrected based on journal requirements. In Table 2, the references were not properly placed in the older manuscript are now corrected with addition of few more references for butyric acid. 

Additional Editor Comments:

1-The phylogenetic tree, Figure 2, number 0.00 and other with decimal must be deleted from the tree. Keep only percentage. In addition, the percentage must be in the right place, not to cross the lines. Branch length no need to give. The legend for the Figure 2 then accordingly must be corrected.

Response: From the phylogenetic tree, i.e., Fig. 1, number 0.00 and other with decimal are deleted. The legend for the Figure 1 is accordingly corrected.

---

## [Editor Report · Decision Letter 2]

16 Jan 2024

Chemo-profiling of Purpureocillium lilacinum and Paecilomyces variotii isolates using GC-MS analysis, and evaluation of their metabolites against M. incognita

PONE-D-23-25925R2

Dear Dr. Lakshman Prasad,

We’re pleased to inform you that your manuscript has been judged scientifically suitable for publication and will be formally accepted for publication once it meets all outstanding technical requirements.

Kind regards,

Ebrahim Shokoohi

Academic Editor

PLOS ONE

Additional Editor Comments (optional):

Almost raised questions was done by the authors
---

## [Editor Report · Acceptance letter]

7 Feb 2024

PONE-D-23-25925R2 

PLOS ONE

Dear Dr. Prasad, 

I'm pleased to inform you that your manuscript has been deemed suitable for publication in PLOS ONE. Congratulations! Your manuscript is now being handed over to our production team.

Kind regards, 

on behalf of

Dr. Ebrahim Shokoohi 

Academic Editor

PLOS ONE